# A Systematic Review of Multifaceted Silence in Social Psychology

**DOI:** 10.3390/bs15091220

**Published:** 2025-09-08

**Authors:** Dat Bao

**Affiliations:** Faculty of Education, Monash University, 19, Ancora Imparo Way, Clayton, VIC 3800, Australia; dat.bao@monash.edu; Tel.: +61-404394073

**Keywords:** silence, choice, culture, power, dialogue, protection, isolation, withdrawal, solitude

## Abstract

This article, conforming to the 2020 PRISMA checklist, presents a systematic review of silence within the realm of social psychology, utilizing research-driven insights. Silence can be interpreted through both interpersonal and intrapersonal lenses; that is, it can originate from external social interactions or be a personal choice. While external silence reflects responses to societal stimuli, internal silence focuses on individual decisions. The piece contends that silence possesses sociological dimensions—when an individual communicates through silence (such as expressing resistance or alienation), they not only convey personal sentiments but may also represent broader collective concerns. Drawing upon the concept of sociological imagination, it posits that what may seem like an individual issue can mirror shared societal struggles, thus highlighting how personal experiences resonate with community dynamics. By examining diverse perspectives of silence, the article elucidates its complexity and significance within social environments.

## 1. Introduction

To explore silence in the contemporary literature and its use, this article undertakes a systematic review of more than 110 texts and seeks to address four central questions: why silence is important to social psychology, what types and functions of silence exist, how culture and context correct our understanding of it and how social psychologists have dealt with silence and what remains wanting. With reference to Mills’ idea of the sociological imagination ([63]), silence is conceptualised not just as something experienced or as silence in itself, but also as a mirror and prism of wider social contexts.

## 2. Why Silence in Social Psychology

To understand why silence holds a vital role in social psychology, this article begins with a concrete example illustrating how non-verbal communication can fulfill a social function. Below is a real vignette designed to make abstract concepts more tangible and relatable.

The incident took place in Kyoto, Japan, where silence often serves as the foundation of social interaction. It highlights the importance of a meaningful moment of silence before engaging in conversation with a stranger. The story revolves around an Englishman named Andrew, who is browsing a supermarket, trying to select some cooking ingredients from the shelf, unsure of which sauces to choose.

As he stands there looking puzzled, a Japanese woman notices his confusion. Although Andrew senses her interest in helping, there is a moment of hesitation on her part, as it is generally uncommon for strangers to initiate verbal exchanges in this context.

Ultimately, her desire to help overcomes her shyness. She approaches Andrew with a warm smile and offers assistance in locating what he needs. Speaking in limited English, she asks him if he is searching for something. To her surprise, Andrew, who has studied Japanese for years, instinctively replies in her language, prompting her to continue the conversation in Japanese.

This delightful encounter illustrates a nuanced social dynamic characterized by mutual observation, exchanged glances, and gestures of assistance, all blended with verbal interactions in two languages. Here, silence is far from empty; it is rich with intention. It embodies a social commitment, as the initial moment of quiet observation created space for the ensuing conversation. Rather than indicating a lack of communication, this silence represented an outward quest for engagement, setting the stage for the interaction that followed.

Without that functional, non-verbal pause, verbal communication could not have emerged. This vignette exemplifies silence’s integral role in social exchanges, highlighting how care and empathy imbue it with a socio-psychological dimension. Countless similar incidents occur daily around the world, often overlooked unless scholars actively seek to illuminate them within academic discourse. This article aims to uncover these meanings through a systematic review of literature, showcasing the diverse functions of silence in our lives.

## 3. The Uncertainty of Silence Research

This article highlights some of the under-theorized knowledge in silence studies. As a scholar who has studied silence for the past 20 years, the author found that little has been done in academic research, especially in social psychology, to adequately investigate this phenomenon other than in a piecemeal fashion. Introduction Silence is omnipresent and critical in human communication and yet it is too often misunderstood, undervalued or de-emphasized relative to articulable speech. To reanimate this, the article posits several zones of scholarly uncertainty. One of these is the fact that the phenomenon of silence has been traditionally under-investigated in the empirical area, even if its relevance has been recognised by academic writers from the 1970s, mostly in the papers related to oral proficiency, and that it has been studied more systematically only from the late 1990s.

There is also a problem surrounding the absence of a singular definition, for silence does not allow of one straightforward definition but is far more fluid in terms of meaning and context when thing about it, which makes the classifying of silence difficult. Moreover, the concept of silence has commonly been approached from an interpersonal perspective, thus focusing less on the intrapersonal aspects such as self-talk, internal discourse and soliloquy.

There are in addition cultural biases reflected in the field since most of the research has been based on one specific cultural context, e.g., Japanese or the Nordic society, without much concern for the diversity of the world, which serves to confine the findings to the peculiarities of a narrow space of applicability. Lastly, it has been upheld by both popular wisdom and so empirical/theoretical academia that silence is often read as the absence (imperfection) of communication and not that it is actively at work, such as in empowerment, resistance, protection or cultural enterprise.

## 4. Focus of the Analysis

This piece scrutinizes the multifaceted interpretations of silence across interpersonal and intrapersonal domains. The former pertains to the influence of societal context on silence, while the latter involves engaging with one’s inner self. Silence contributes multiple narratives that often go unheard unless there is a willingness to understand its origins, implications, and connections. The article investigates how societal forces shape these narratives and the psychological frameworks within which silence manifests. Silence serves numerous roles, reflecting both personal introspection and broader social commentary.

Wolfgang Amadeus Mozart aptly stated that “silence at the right moment is golden,” a sentiment that is relevant not only to music but also to effective social communication. Even in its apparent absence, silence plays a vital role in social psychology. While silence is frequently misinterpreted as a void in communication, it carries various significant connotations that warrant attention. This article aims to delve into the rich dimensions of silence, illuminating its psychological ramifications and the socio-cultural contexts that shape our understanding.

## 5. Methodology

This review employs a textual research framework that unfolds in two key stages. Initially, 400 academic texts concerning silence were chosen from a wide range of databases, including PubMed, Scopus, Web of Science, JSTOR, Academic Search Complete, ScienceDirect, Google Scholar, Cochrane Library, PsycINFO, ERIC, ProQuest, IEEE Xplore, and SAGE Journals. The search lasted from March 2024 to April 2025. The search was conducted manually and intuitively without specifically identifiable tools or strategies. The selection, based on five criteria: the inclusion of silence studies (a) empirical research, (b) historical perspectives, (c) recent developments, (d) relevance to current discussions, and (e) comprehensive debates. As a result, 111 items were selected for the review.

In the second stage, these texts were subsequently analyzed in a second wave using four key strategies: (a) content analysis, wherein patterns over the surface of the material are recognized ([53]); (b) discourse analysis, in which attention is paid to how language reflects ideology and power relations ([32]); (c) hermeneutic interpretation, based on traditions of meaning-making through engaging textually ([35]); and (d) close reading, which involves extracting information at the level of detail.

The initial search yielded 400 records across multiple academic databases, including Scopus, JSTOR, PsycINFO, and Google Scholar, spanning research published from the 1970s to 2025. After removing duplicates and non-English texts, 342 articles remained for preliminary screening. Titles and abstracts were reviewed for relevance to the social psychology of silence, leading to the exclusion of 189 records. Full texts of the remaining 153 articles were assessed using predefined inclusion criteria: relevance to silence as a psychological or sociocultural construct, conceptual clarity, and contextual diversity. Ultimately, 111 studies were included in the final synthesis.

The researcher carried out the systematic review in several stages to ensure this process would be as transparent and rigorous as possible. In the first step, all types of records found through database searching were combined, and duplicates were deleted. The studies that were left were then screened on titles and abstracts to delete those studies that seemed irrelevant to silence in social psychology. In the second stage, full-text articles were reviewed for relevance, methodological integrity, and conceptual relevance to the inclusion and exclusion criteria. Studies that did not contribute empirical or theoretical insights were not included. This process led to the final range of publications that were included in the review as outlined in the flow diagram (Figure 1).

To enhance methodological rigor, the review incorporated not only a systematic screening process but also a critical appraisal of study quality. Each of the 111 included studies was assessed against three dimensions: methodological integrity (clarity of design, sample, and data collection), conceptual clarity (how silence was defined and operationalized), and contextual diversity (the extent to which the study accounted for cultural or situational factors). While not every study scored equally across these dimensions, the process ensured that the synthesis moved beyond mere description toward evaluative comparison. By explicitly signalling the strengths and weaknesses of the included evidence, this appraisal contributes to a more transparent and trustworthy review.

This review was retrospectively registered on PROSPERO under the identifier CRD420251102598. The registration record provides full details of the protocol, including eligibility criteria, search strategy, and planned synthesis methods, ensuring transparency and reproducibility.

The review highlights silence as a multidimensional construct that transcends cultural, psychological, and communicative boundaries. The included studies show that silence can be both deliberate and unconscious, functioning as a form of resistance, contemplation, politeness, marginalization, or emotional regulation. A dominant theme across the literature is the cultural relativity of silence, with varying norms and interpretations found across East Asian, Western, and Indigenous contexts. For instance, what may be perceived as passivity in one context may represent thoughtfulness or deference in another.

## 6. Cultural Perspectives on Silence

Despite its pervasive nature, silence often carries negative connotations in both mass media and public discussions ([73]). Prominent speakers frequently overlook the constructive role of silence, framing it instead as detrimental. For instance, an online search for “silence” reveals that many discussions link silence to serious issues like danger or trauma, with very few celebrating it as a source of inspiration or learning.

In educational settings, while teachers occasionally acknowledge the value of quiet students—suggesting a shift toward recognizing their diligence—this remains an exception rather than the rule. Research highlights the notion that silent students can be intellectually engaged, yet societal perceptions still often undervalue the silent voice ([29]; [86]; [24]; [87]).

## 7. The Study of Silence in Psychology

Research interest in silence has not been confined to psychology alone ([74]; [47]; [96]; [28]); it extends across various academic disciplines. Tracing back to the 19th century, early inquiries into silence originated within the fields of psychology and philosophy, progressively weaving into education and communication studies since the 1970s. Although the term ‘silence’ has only recently gained traction within scholarly discussions, the groundwork laid by previous researchers allowed for an exploration of new dimensions, including reflection and engagement without sound ([62]; [44]; [72]; [45]; [14]; [70]; [102]).

The late 20th and early 21st centuries saw silence being redefined beyond mere absence of speech, contributing to a deeper understanding of its varied implications. Scholars proposed classifications such as psycholinguistic silence—characterized by hesitations that enrich communication—interactive silence, and sociocultural silence, each emphasizing the distinct roles silence plays across different contexts. By tapping into these diverse dimensions, silence emerges as a critical theme worthy of continued exploration across all areas of human interaction and communication ([76]; [5]; [18]; [23]; [71]; [46]).

The 1980s marked a growing recognition of the complex meanings associated with silence in speech communication ([91]). Silence was frequently interpreted not just as absence but also as an indication of refusal to engage ([100]) or as a mechanism for control and resistance ([38]). However, from the 1970s to the 1990s, empirical studies exploring the role of silence in communication were notably scarce ([46]; [26]).

Despite ongoing calls for research, the significance of silence in educational contexts often faded into obscurity during this period. In contemporary discussions, silence does not solely imply total quietness but can also represent minimal talk within classroom interactions ([79]). In cultures that value silence, it is seen as equally valuable as speaking, as it allows for reflection on what has been conveyed ([106]). Silence conveys respect, promotes harmony, and fosters ‘attentive listening and active thinking’ ([58]). Sometimes, silence itself can serve as a form of expression. At the same time, spoken words may be classified as external or interactive speech ([80]; [81]), silence can represent a stage for internal dialogue, rehearsal, sub-vocal articulation ([78]), and the assimilation of speech patterns ([65]).

Recent scholarly endeavors have aimed to examine the interplay between silence and spoken communication in a more nuanced manner than simply categorizing them as noise versus silence. Engaging with silence in educational discourse involves grappling with a rich tapestry of voices. Studies have begun to classify silence into various contexts and functions—such as affirmation, debate, informal conversation, formal lectures, negotiation, critique, inquiry, and denial. Just as spoken dialogue can be meaningful or irrelevant, silence can also vary in significance, with categories including meaningful silence and trivial silence. If conversation that reflects educated discourse is viewed as “the speech of educated individuals” ([31]; [84]), then it follows that educated silence can be defined similarly.

The call for recognizing silence within education has persisted for seventy years. However, it is only in the past two decades that substantial research has started to engage with this concept meaningfully. While earlier calls from the 1950s to the 1990s emphasized silence as a mechanism for meaning making ([76]; [26]; [84]; [27]; [48]), the push for silence research since the 2000s has gained momentum, driven by concrete suggestions for exploring social communication.

Over the last twenty years, the discourse surrounding silence has sought a deeper understanding of its implications. Recent scholarship has contributed insights into themes such as misunderstandings surrounding silence in communication ([34]; [83]), the reinterpretation of silent behaviors across diverse cultural settings ([40]; [69]), feelings of frustration linked to silence in cross-cultural interactions ([52]; [97]; [67]), the negotiation of multiple identities ([66]), the interpretation of meanings derived from silent encounters ([7]; [41]; [52]), and the adaptability of behavior ([52]; [108]; [88]; [104]).

## 8. Defining Silence

Defining silence is one of the most intricate challenges within the study of non-verbal communication. Given the extensive debates surrounding its meaning, establishing a singular definition proves elusive. Silence is inherently fluid, possessing comparative rather than absolute meanings. While it literally signifies the absence of sound or speech, researchers have recognized that silence can encompass both sound and speech. For instance, one individual may remain silent while others are conversing, leading to overlapping expressions of words and silence.

The term ‘silent’ is often relative, not implying total muteness ([8]). Wilhelm Wundt, a German psychologist and philosopher, posited that language comprises two dimensions: the internal and the external. Silence, deeply linked with social discourse, represents an interactive process that responds to the behavior of others ([13]). Given the inner dynamics guiding social communication, focusing solely on the spoken word only scratches the surface of understanding how we interact ([11], [12]).

Perceptions of self-vocalization vary significantly; one person may consider their speaking patterns moderate while others perceive them as overly verbose or insufficiently engaging. Silence plays a key role in the intricate dialogue between listener and speaker ([6]; [16]). While actively listening, the audience not only seeks comprehension but also engages in an internal conversation with the shared content. Furthermore, conversational partners may influence each other’s engagement levels ([15]). A skilled facilitator, for example, can help a reserved participant express themselves more freely compared to someone who lacks such skills.

In online environments, such as chat forums or email communications devoid of audible speech, silence can manifest as the absence of contributions in the shared digital space ([15]). In this context, our understanding of silence must evolve; it should not hinge solely on the absence of vocalization. In these virtual settings, silence can signify non-participation or an opportunity for reflection before engaging in dialogue.

Some scholars conceptualize silence through its cognitive functions. For instance, in De Guerrero’s empirical research ([28]), silence is associated with seven characteristics: ideational (thought creation), mnemonic (memory retrieval), textual (text structuring), instructional (application of linguistic rules), evaluative (self-monitoring and correction), interpersonal (visualization of dialogue with others), and intrapersonal (inner speech practice). Other theorists connect silence to additional elements like communication and gestures, referring to it as eloquent silence—intended to convey meaning ([2]) or semiotic silence, accompanied by visual cues such as gestures, facial expressions, or culturally significant artifacts that function as ‘silent proverbs’ requiring specific cultural knowledge to interpret and decode ([2]; [103]).

Some researchers establish a dichotomy in silence, distinguishing between active and passive forms (also known as busy versus idle silence ([50]). Active silence occurs when individuals consciously choose not to speak, whereas passive silence arises from inability or restriction. Another framework distinguishes between weak and strong silence, where weak silence may serve as a punitive measure for behavioral issues, while intense silence facilitates personal space for exploration and learning ([20]). Others redefine silence entirely by excluding the auditory component, asserting it is not merely about sound absence but about a lack of shared ideas.

Thus, in today’s digital communication landscape, silence is the absence of written contributions rather than vocal expressions. Terms associated with interaction have evolved into digital connotations, reflecting how silence transforms with changing modes of communication. As concepts of social presence and engagement have shifted ([39]; [57]), so too has silence come to signify quietude in writing rather than in speech ([107]).

The multitude of perspectives on silence reveals its complexity, rendering it challenging to adhere to a single definition. Some scholars strive for simplicity in describing silence, seeking alternative terminology that encapsulates its diverse roles. Each term reflects a distinct perspective on silence, demonstrating its potential as a nuanced tool used for various communicative purposes.

## 9. Silence with Interpersonal Significance

### 9.1. Silence as an Element of Human Nature

Humans thrive on routines, and while some individuals revel in social activities, others find solace in a harmonious blend of solitude and companionship ([14]). We are inherently private beings, often hesitant to share our innermost thoughts and emotions for various valid reasons. Just as we require physical space away from crowds to recharge, we also need mental breaks from the constant chatter of others to rejuvenate our minds and foster new ideas ([9]).

In fact, we typically spend more time in silent contemplation—while mentally engaged—than we do vocalizing our thoughts. Silence, often perceived as the mere absence of sound, is closely intertwined with spoken words. Even during moments of silence, an inner dialogue frequently unfolds within us ([92]; [10]). As [26] ([26]) notes, “voice is not independent of silence.” This interconnectedness makes silence a complex construct that cannot easily be isolated or defined.

Silence has profound interpersonal implications. For instance, individuals attempting to navigate a new culture might feel intimidated by the gaze of their peers and subsequently withdraw. Several studies by [9] ([9], [10]) recount experiences in which Japanese individuals, seeking to adapt to Australian culture, succumbed to shyness when aware of fellow Japanese observing them, leading to hesitation in fully embracing the new environment.

Moreover, silence is highly contextual. The degree to which individuals utilize silence is influenced by personality traits, sociocultural settings, and the mental demands of the communication at hand ([17]). Those who are verbally expressive in their native language can often find themselves unusually quiet when conversing in a foreign tongue ([7]). Silence, like spoken language, is not devoid of context ([11]); it requires an awareness of when and how to be silent to ensure acceptance, respect, and clarity in communication.

### 9.2. Silence as a Personal Choice

Research has demonstrated that silence can be a conscious decision made by individuals. Some may choose silence as a means of exerting self-control, exercising discipline ([14]), or exercising caution ([43]; [36]). For others, remaining quiet serves as a protective mechanism, particularly in situations where sharing one’s thoughts might lead to undesirable consequences. As noted in a study by [85] ([85]), the Japanese often opt for silence as a haven amid conflicting cultural influences. Similarly, [95]’s ([95]) research on Japanese students abroad highlights that many remain silent due to the challenge of expressing their true identities.

A study by [60] ([60]) reveals that children frequently use non-verbal methods to communicate their thoughts and preferences, demonstrating their capacity for agency in early educational settings. Choosing silence enables individuals to maintain their privacy or sidestep potential conflicts. In specific contexts, silence may also serve as a strategic tool to convey dissent or lack of agreement, compelling others to acknowledge their presence or viewpoint from a different angle.

### 9.3. Silence as Cultural Expression

Silence carries cultural significance, with differing societies ascribing various meanings and values to it. In the 2019 BBC series Duty/Shame, a scene unfolds in a café between a Japanese man named Kenzo and an Irish woman named Sarah. After a brief exchange about family, Sarah breaks the silence with a personal question, leading Kenzo to inquire why she feels discomfort with silence. This exchange illustrates that silence can hold disparate meanings across cultural contexts; what is considered normal in one culture may be perceived as unusual in another.

In Japanese culture, silence functions as a means to maintain social harmony, evading confrontation or showcasing genuine feelings ([41]). In Nordic nations such as Sweden, Finland, and Norway, silence is valued as an integral part of communication, with individuals often assessed by their actions rather than their verbal output ([93]). For many Native Americans, silence serves as a protective barrier against dominant narratives, embodying resilience rather than timidity ([82]). Among the Amish, silence is a cornerstone of their simple, faith-centered lifestyle, allowing space for reflection and community meditation ([21]). Similarly, in Tibetan Buddhism, silence is revered as a path to inner peace, with practices like silent meditation being central to spiritual growth ([64]).

Regarding personality, silence may indicate humility. For some, minimal speech reflects refined manners, while excessive talk might be perceived as arrogance ([59]). In cultures such as Japan and Turkey, silence often embodies modesty, whereas excessive verbal expression can denote superficiality. A prudent approach to speaking is usually valued, helping individuals avoid being intrusive, irrelevant, or rude. Empirical research by [7] ([7]) and [69] ([69]) on Japanese communication styles, along with studies by [77] ([77]). On Chinese communication behaviors, it is suggested that respect and self-restraint essentially inform their silences. People in these cultures tend to keep a low profile to mitigate peer pressure and conflict.

Conversely, other cultures prioritize verbal expression over silence. In North America, cultures such as the United States and Canada celebrate open dialogue, valuing freedom of speech. George Washington famously remarked in 1783, “The freedom of speech may be taken away, and, dumb and silent we may be led, like sheep, to the slaughter.” Many Latin American societies, known for their passionate communication styles, value conversation as a fundamental means of connection. Mediterranean countries like Greece, Italy, and Spain often favor spirited discussions in which expressive speech fosters relationships. Likewise, Indian culture places a premium on eloquence, particularly through storytelling, debates, and intellectual exchanges, signifying wisdom and cultural appreciation. Many African traditions are steeped in oral history, where storytelling, proverbs, and engaging dialogue are vital to cultural identity, leading to the integration of public speaking training as a key aspect of personal development ([3]).

### 9.4. Silence as Empowerment

[1] ([1]) posits that silence can serve as a metaphor for power, signifying the ability to influence others. This concept has inspired filmmakers to utilize silence as a creative device to elicit emotional responses from audiences. Moments devoid of sound often intensify the viewers’ empathy, as when a character’s quiet anxiety takes center stage. Silence can create a void that invites the audience to engage their imagination and emotions. Brazilian French filmmaker Alberto Cavalcanti famously articulated, “Silence can be the loudest of noises, just as black, in a brilliant design, can be the brightest of colors.”

In conflict situations, silence can serve as a strategic maneuver, confusing opponents and compelling them to guess the next move. This uncertainty can instigate anxiety and vulnerability in others. Sometimes, the most impactful characters are those who speak little, while less formidable adversaries may dominate the conversation. In Jean-Jacques Annaud’s film Enemy at the Gates, the lead character Vassili Zaitsev conveys determination and intelligence through potent silence and close-up shots that reveal his resolve. The tension in silent moments emphasizes the confrontation, underscoring how his quietness adds depth to the suspense. [1] ([1]) further argues that power is not a tangible entity, but rather arises from dynamic relational processes.

In everyday situations, wielding silence effectively can be a powerful influence tactic. Opting for silence during discussions can create suspense and amplify the impact of forthcoming words. Well-timed pauses can make statements resonate more deeply, ensuring they leave a lasting impression. Research indicates that influential silence—utilized by leaders and authoritative figures—can amplify the effectiveness of their statements. Additionally, silence plays a critical role in managing impressions; individuals may strategically use pauses to enhance persuasiveness, credibility, and perceived knowledge. When silence carries a complicated subtext, it can become potent and unsettling to others ([19]).

### 9.5. Silence as Powerlessness

When silence signifies indifference—such as neglect or lack of attention—it often leads to feelings of powerlessness. Research conducted by [22] ([22]) indicates that adolescents frequently experience unhappiness and stress when they feel overlooked by their peers and family. Such neglect, characterized by judgment and disengagement, can result in loneliness and diminished self-worth, ultimately impacting a person’s identity and life trajectory. This phenomenon is closely linked to social anxiety. For instance, [105] ([105]) discovered that social anxiety can manifest in psychosomatic symptoms, including headaches, stomachaches, sleep disturbances, and excessive worry, which may culminate in school refusal, isolation, and depression. The case of Roma minorities in [75]’s ([75]) study illustrates how the dominant society often denies individuals facing prejudice their humanity. They not only lack a voice but are also discouraged from embracing their true selves.

Schools are not just places of learning; they can also serve as political arenas where power dynamics shift, and voices can either be cultivated or stifled. Such dynamics can lead to social inclusion or exclusion, impacting individual dignity. For example, an introverted student who enjoys thoughtful reflection might feel marginalized when pressured to speak, while an outgoing student who wishes to share their thoughts may be ignored. These experiences can foster feelings of isolation and alienation within the community.

Educators wield significant influence in this dynamic. As evidenced in the research by [4] ([4]) as well as [54] ([54]), teachers often hold the power to dictate who speaks, who remains silent, whose voice is amplified, and who is silenced. When faced with oppression or unfairness, some assertive individuals may resist, while those who feel vulnerable might succumb to becoming subjugated. [95]’s ([95]) research indicates that self-inhibition and lack of self-expression often lead to undesirable silence. Those experiencing this phenomenon may retreat behind a barrier of silence marked by shyness, embarrassment, and internalized shame, as highlighted in [89]’s ([89]) study of Japanese students. [42] ([42]) also suggests that allowing written expression might be beneficial for students who are hesitant to verbalize their thoughts. In line with this, [85] ([85]) recommend building supportive social networks to boost self-esteem and combat depression.

## 10. Silence with Intrapersonal Significance

### 10.1. Silence as Self-Dialogue

Plato recognized over 2300 years ago that thought can be viewed as a form of internal dialogue—a conversation with one’s own soul. This notion influenced early research on inner speech, particularly through Vygotsky’s theories during the 1960s. Empirical studies, including those conducted by [87] ([87]) and others ([28]; [47]) have identified the role of inner speech in processes like reading comprehension, language acquisition, cognitive processing, and learning experiences. A crucial point in this body of work is the understanding that, when engaged in listening, learners not only absorb external speech but also participate in their own mental dialogue. At times, attention shifts from the speaker’s words to the learner’s internal reflections, which can be facilitated by what has been termed the ‘psychic ear’ ([37])—a transition from external voices to one’s own internal narrative.

Psychological research has shown that personality traits significantly influence communication styles. Extroverts thrive on exchange and interaction, while introverts typically prefer to listen, reflect, and respond only when it feels necessary ([25]). For many, this internal whispering—identified as inner speech or self-directed thought ([29]; [33])—is a vital part of their daily experience.

Although this internal silence may not take on linguistic forms, it contains thoughts devoid of explicit words ([98]). When thoughts become articulated, they can evolve into what is termed “private speech” ([55]). This fluidity allows for the transition from abstract musings to concrete verbal expressions, facilitating the process of translating thoughts into spoken words. While inner speech may not always reach our conscious awareness, it can become vocal when there is a desire to share one’s ideas. [92] ([92]) emphasizes that the connection between inner and private speech can lead to external articulation, should one choose to express their reflections.

[51] ([51]) highlights instances where students resist engaging with standard, predetermined educational content. Instead, they seek to construct their own understanding, aligning it with their personal goals and needs. This self-dialogue empowers students to discover new insights and knowledge on their own terms.

### 10.2. Silence as a Means of Solitary Enjoyment

In recent years, awareness surrounding the benefits of silence has surged, particularly due to the growing interest in mindfulness and meditation practices in educational contexts ([56]). Silence provides an opportunity to delve into one’s inner self, uncovering undiscovered potentials ([99]). It serves as a space for appreciating solitude. For instance, [30]’s ([30]) case study illustrates how individuals can learn to embrace and cope with being alone, while [101] ([101]) explores children’s experiences with guided reading during the pandemic, highlighting how moments of isolation can foster enjoyable, meaningful engagement.

Silence can also represent comfortable solitude. [68] ([68]) notes that while the digital age enables instant connectivity, it paradoxically fosters loneliness and complicated relationships. Even with readily available communication, many individuals protect their personal space and establish boundaries to avert invasion, leading to isolation.

Research by [22] ([22]) outlines the crucial role solitude plays in the lives of adolescents. Alone time offers a safe and valuable space for various activities, and how adolescents utilize this solitude can significantly influence their well-being. Interestingly, while solitude has traditionally been viewed negatively in adolescence, with a focus on loneliness and peer exclusion, it is now recognized as a vital aspect of normal development.

Silence can manifest as an individual experience or a collective phenomenon. While it is deeply personal, there are moments of shared silence that foster mutual understanding. When two individuals enjoy a quiet moment together, it can signify empathy or connection. In larger groups, collective silence may communicate a shared attitude, whether that be resistance or acceptance. Societal values dictate the role of silence, with some cultures practicing silence similarly to how others engage in speech. Silence can also have social or personal dimensions, being social when one refrains from speaking to avoid trivial conversation, or personal when one seeks mental clarity and peace. While silence can indicate mindfulness when actively used for contemplation, it can also signify a need to disengage from persistent thoughts.

### 10.3. Silence as Social Withdrawal

In Tove Jansson’s fairytale “The Invisible Child,” the protagonist Ninny is rendered silent and invisible due to her mistreatment, with silence serving as a form of resistance. When someone is quiet for an extended period, they may become invisible ([4]). If silence is imposed upon individuals, it can lead them to adopt invisibility as a protective measure.

Despite efforts by many societies to champion local languages and cultural diversity, various communities continue to face challenges in achieving social integration. The Roma and Gypsy populations in Europe serve as poignant examples of marginalized groups characterized by their nomadic identities and a longing for belonging ([75]). As a result, these minorities often experience prejudice and marginalization, silenced and unsupported. They may quit openly expressing their concerns in public, leading to what is known as self-silencing. This silence, particularly among the powerless, can damage mental health and well-being. It is associated with feelings of frustration and a diminished sense of autonomy, with evidence suggesting that self-silencing can heighten depression and anxiety, reducing overall psychological health. Thus, silence can stem from oppression, ignorance, or serve as an assertion of power.

Anxiety can also drive individuals to withdraw socially, particularly in interpersonal interactions ([61]). Many find solace in online communication, using it as a safe space to express themselves. [49]’s ([49]) study reveals that individuals with anxiety often engage more actively in online forums, where they can carefully consider and refine their statements. Similarly, [94]’s ([94]) case study chronicles Daniela, an adult immigrant from Mexico, who grapples with anxiety, language barriers, social isolation, power imbalances, and economic challenges as she navigates her new environment in the United States.

Though these views on silence may appear to run counter, they are linked by the same foundation of attention towards how silence is inundated with relational significance. Silence, socially and psychosocially, is a relational act, that can either grant dignity or violate it. But in the hands of writers, silence can also serve as an intentional communicator of care, respect or forgiveness that opens up spaces of empathy and recognition ([54]). But when the silence is forced or born from fear, shame, or exclusion, that absence of words can compound alienation and cause harm. In so doing, silence is not inherently positive or negative in and of itself: it is an issue of relationality, and meaning depends on both the circumstances, power, and the quality of the human relationship. In holding these dualities together, we also see silence as not a pair of opposites but a spectrum of human experience—able to envelop both kindness and trauma, with countless gray shades in between.

## 11. Silence as an Expression of Kindness

Kindness manifests both in our interactions with others and in how we treat ourselves. It encompasses treating others with goodwill and showing compassion toward the self. Our approach to kindness includes friendliness and respect, which can be reflected through silence. Rather than viewing silence negatively, we can harness it productively to foster stronger communities. Silence can signify a stand against social injustice, embody tolerance, or communicate forgiveness—indeed, it can express more profound truths.

### 11.1. The Art of Truthfulness

Silence can also serve as a mechanism for confronting social inequality. In [90]’s ([90]) research involving a drama company named Shizuoka No-Borders, performers utilized gestures and movements devoid of spoken words. In this performance, distinctions between individuals with disabilities and their able-bodied counterparts were minimized; all participants were recognized simply as performers. This approach redirected the audience’s focus from labels to the humanity of individuals, promoting an understanding of diversity rather than categorization.

### 11.2. Tolerance and Forgiveness

Silence often embodies forgiveness. Audiences witnessing the drama above may feel sympathy toward performers with disabilities, resulting in an atmosphere tinged with discomfort that discourages laughter. Yet, this very silence can also create space for forgiveness. Viewers may feel invited to respond humorously to the scene, experiencing equality in enjoyment irrespective of disabilities. The performance was crafted to embrace inclusivity, encouraging laughter among all participants—both those with disabilities and those without.

The present review also seeks to consolidate the multiple frameworks previously applied to silence studies into a more coherent theoretical anchor. Drawing together sociological imagination, sociocultural theory, and communicative pragmatics, the re-view advances the notion of silence as spectrum: a continuum that spans intrapersonal reflection, interpersonal negotiation, and sociopolitical dynamics. This integrative model helps explain how silence can simultaneously function as agency, resistance, vulnerability, and care. While modest in scope, this conceptual consolidation strengthens the theoretical contribution of the paper by moving beyond fragmented frameworks toward a synthesized perspective.

## 12. Limitations of the Study

Two notable limitations of this textual research approach should be acknowledged. First, each selected article reflects the unique perspective of its author, and thus may not encapsulate a comprehensive view of silence studies across the academic community. To mitigate potential bias, the author has made an effort to integrate and contrast varying viewpoints. For example, the silence of the Roma minority was analyzed not in isolation but in relation to the dynamics of the dominant group. When a minority voice is suppressed, it often leads to social withdrawal, reinforcing the complex interplay between silence and sociopolitical power.

Second, the articles reviewed focus on specific cultural contexts, which may limit their applicability to broader social settings. To address this concern, this study contextualizes each interpretation of silence. For instance, when silence is viewed positively, it is often framed within Japanese or Nordic cultures, while its negative connotations may be examined in North American or Latin American contexts. This approach ensures that readers understand the meanings of silence embedded within relevant sociocultural frameworks rather than viewing them in isolation.

The review process also contains several limitations within the included body of evidence that merit attention. There is an overrepresentation of studies from East Asian and Western contexts, which may limit the generalizability of conclusions to other global regions such as Africa, the Middle East, or Latin America. Moreover, many studies rely on qualitative self-report data, which, while rich in detail, can introduce subjectivity and may lack longitudinal insight. The review reveals that silence is not merely the absence of speech but a psychologically, culturally, and socially rich phenomenon. Across the reviewed literature, silence emerges as both a communicative act and a strategic response to sociocultural dynamics. The findings align with prior research that highlights silence’s role in power negotiation, identity construction, and interpersonal relationships.

A final limitation is that the present review does not propose a fully developed conceptual model, but rather lays groundwork for such a framework. Future research could test the spectrum model empirically, for example, by operationalizing silence as both an intrapersonal and interpersonal construct across cultural contexts. By acknowledging this limitation openly, the study emphasizes its role as a foundation for subsequent theoretical development rather than a definitive endpoint.

## 13. Implications and Future Directions

The discussions presented in this article are grounded in contemporary research rather than merely theoretical assertions. It reflects a balance between historical perspectives on silence and the latest insights in the field. Society must deepen its understanding of silence, which is as significant to individual experiences as it is to communal life. Influenced by collective dynamics and sociopsychological factors, silence is laden with hidden voices yearning to be acknowledged. Its meanings are diverse, and the potential for further exploration is vast.

What has been explored here represents only a fraction of what remains to be uncovered. Our understanding of silence is rife with blind spots, waiting to be explored. The *Journal of Silence Studies in Education*, for instance, founded in Australia in 2021 by experienced scholars in this area, has enumerated eighty distinct themes for future research, hinting at the need for even more exploration. Silence encompasses a complex landscape that is academic, linguistic, sociocultural, psychological, political, and communicative. In contrast to speech, silence often eludes observation, necessitating a shift in focus toward the unarticulated dimensions of experience that qualitative research sometimes overlooks.

Despite the extensive efforts made in this article to illuminate various interpretations of silence, gaps in our comprehension persist, keeping the field open for further inquiry. Silence, like speech, manifests in context; examining it in its environment is crucial. The shifting surroundings will ultimately shape the expression and subsequent meanings of silence. Therefore, silence only gains clarity when we examine both its origins and its impacts. In essence, silence proves to be more tangible than we often realize, with identifiable triggers and outcomes. The outcome of the search is summarized in the bar chart below (Figure 2).

To complement the qualitative synthesis, a series of quantitative mappings was conducted. The distribution of studies across time shows a noticeable growth in silence research from the early 2000s onward, reflecting a shift from sporadic inquiry to sustained academic interest. Geographically, the strongest representation came from East Asia (36%), followed by Europe (28%) and North America (21%), with comparatively limited attention to Africa, Latin America, and the Middle East. In terms of re-search design, 62% were qualitative (case studies, interviews, ethnographies), 24% mixed-methods, and only 14% purely quantitative. These descriptive statistics help illuminate disciplinary trends, highlighting both concentrations of knowledge and underexplored gaps.

The bar chart below illustrates the article’s emphasis on various themes, with cultural perspectives receiving the most extensive coverage, reflecting a strong focus on how silence functions across diverse societies. Interpersonal and intrapersonal silence are also prominently featured, exploring silence both in social interactions and as a form of internal reflection. The article draws a compelling contrast between silence as a form of empowerment and as a manifestation of marginalization, particularly within educational and minority contexts. Methodological discussion is given notable attention, outlining the two-phase review process and qualitative synthesis approach. Educational settings, digital and online silence, and future implications are also covered, though to a lesser extent. Overall, the chart encapsulates the article’s multidimensional exploration of silence, highlighting its psychological, cultural, and communicative complexities.

## 14. Summary of Main Findings

Based on the systematic analysis of 111 studies, we find that silence in social psychology is a multifaceted and complex variable in terms of its communicative and psychological meaning. Several themes emerge as follows:

Inter- and intrapersonal functions: There are two levels of operation of silence. On an interpersonal level, it governs relationships, signifying care, respect, bargaining, hesitancy, defiance, or retreat. And in interpersonal terms, it mediates inner voice, whispering, brooding, and solitary indulgence. These results suggest that silence can be considered not only a social signal, but also a cognitive and emotional accessible resource.

Cultural relativity: The meanings attached to silence vary enormously in different cultural environments. The Matrix of Silence and Speech Over the past decade and a half, researchers have been exploring the relationship of silence to speech and voice in various cultural settings around the world. In East Asia and the Nordic societies, silence is often used to signify attentiveness or harmony; in North America and Latin America, silence is most often read as a lack of voice or disengagement. This has implications for policy related to education and communication, which should acknowledge silence as a culturally specific rather than universally determined entity.

Ambiguous social emblems: Silence as power and as weakness. It may be a tactic of emphasis, of governing, of refusal, but it is also symptomatic of marginalization, exclusion, trauma. For practice, this underlines the importance of attending to the relational context: silence that preserves shame in one context may perpetuate alienation in another.

Educational and clinical significance: Silence is characterized as contributing to reflection and attentive listening in classrooms, but appears to be less recognized by teachers. In clinical or therapeutic contexts, the silence may enhance a relationship, offer protection, or permit self-disclosure. These results highlight the importance of professional development that posits silence as an active form of participation rather than an absence of communication.

Conceptual and methodological inadequacies: Research on silence remains a field that is not homogeneous itself. It is also inconsistently conceptualized; predominantly focused on East Asia and the West, and does not adequately consider intrapersonal or digital silence. The pre-eminence of qualitative self-report data restricts the exploration of trends over time. Future studies could benefit from a more interdisciplinary approach, incorporating a wider geographic sample and paying closer attention to online and intercultural communication.

Collectively, the results render silence as a psycholinguistic space, a sociocultural practice, and a psychosocial marker of agency and disempowerment. The review also highlights the clear implications of the findings in various areas. From an educational perspective, acknowledging silence as a worthy mode of participation could foster more inclusive classroom practices that respect learners’ preferences. For policymakers, the results suggest that there is a need for cross-cultural communication protocols that can tolerate silence as a cultural and communicative norm. In therapeutic settings, silence should be considered not only as a lack of response but as an active aspect of the relational experience. For research: More cross-disciplinary and context-referred research, particularly in understudied regions and digital environments, is encouraged within the review. In general, the evidence indicates a movement away from treating silence as nothing, to acknowledging it as an active and productive way of participating that affects learning, interaction, and health.

## 15. Significance of the Study

This research is necessary because it integrates a fragmented literature to show that, contrary to common belief, silence is not a peripheral or leftover part of communication but a multilevel phenomenon at the heart of social psychology. Drawing on evidence from more than a hundred studies, the review recasts silence as a communicative and psychological resource with applications spanning cultural, educational, and clinical contexts. It demonstrates the dual nature of silence as both an empowerment and a vulnerability, as a solitary self-reflective act, as a social negotiation, and as indicative of cultural identity and marginalisation. In pursuing such aims, the study affords better conceptual clarity to further research, invites more socially and culturally responsive pedagogic and therapeutic practices, and assists in informing policy making that recognises silence as a valid and creative form of participation.

## 16. Conclusions

Silence holds hidden voices yearning to be fully acknowledged, yet much of its potential remains underexplored. The review has identified a modest range of themes that invite further research. In this context, silence has been recognised as a psycholinguistic space, such as pauses for thought; a sociocultural function, operating as a norm in particular cultural settings; an empowering tool, employed for emphasis or resistance; and a sign of powerlessness, often linked to marginalisation or trauma. It is also understood as self-dialogue, a form of intrapersonal communication, as well as an expression of the need for solitude, offering opportunities for mental rest and mindfulness. Based on these premises, the article delivers a call to action for more inclusive, interdisciplinary, and context-sensitive studies on silence in social psychology. Based on the empirical gap after analyzing over 100 texts, the article calls for more nuanced, multidisciplinary research to understand the diverse meanings and impacts of silence.

This review adds to the existing literature theoretically by considering silence not as a marginal or leftover category of communication, but rather as a multidimensional construct cutting across cognitive, sociocultural, and psychological domains. By comparing insights from more than 100 studies, this chapter adds to current understanding. It illustrates how silence operates at once as an intrapersonal resource (for reflection and self-talk), an interpersonal strategy (for being caring or resistant or negotiative), and a sociopolitical marker (of power and exclusion and trauma). These theoretically driven findings have practical implications to indicate that educators, health professionals, and individuals in policy roles need to reconceptualize silence as an active and productive state of participation, as opposed to an absence. Further study could extend this synthesis to an examination of how silence functions within digital spaces, with marginalized groupings, and among the interculturally diverse to facilitate theoretical grounding for future work on understanding communication complexity.

## Figures and Tables

**Figure 1 behavsci-15-01220-f001:**
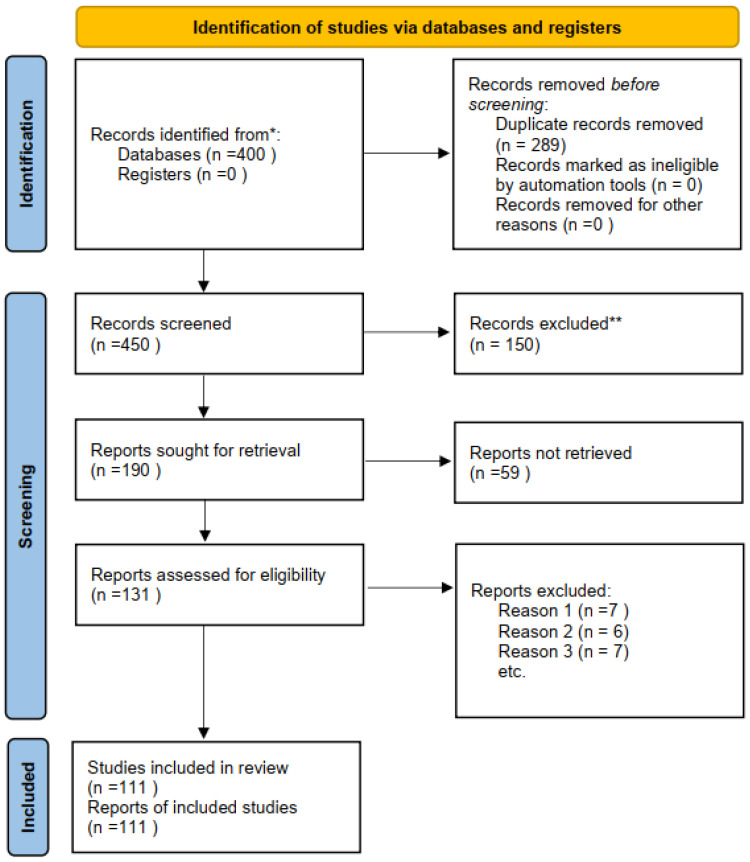
The Screening Process of Academic Resources. (*) Indicates reasons for exclusion at the screening stage (e.g., not peer-reviewed, not relevant to the research focus). (**) Indicates clarification of record numbers after duplicates were removed.

**Figure 2 behavsci-15-01220-f002:**
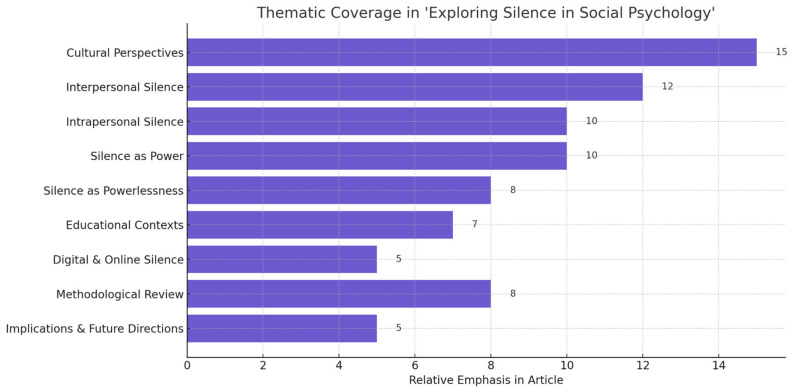
Thematic Coverage in ‘Exploring Silence in Social Psychology’.

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
