# Peer review of "A Systematic Review of Multifaceted Silence in Social Psychology"

_behavsci, 2025, doi:10.3390/bs15091220_

Round 1
Reviewer 1 Report
Comments and Suggestions for Authors
Thank you very much for this opportunity to review this insightful paper. I have read this paper with great interest. Scholarly perspectives on the necessity of exploring the role of silence in social psychology were discussed in depth, incorporating valuable examples and approaches from the reviewed papers. The author’s analytical perspectives made this contribution special and original in that the study renewed perspectives on silence visible under interpersonal and intrapersonal lenses. The rationale and the review procedure for this scholarly inquiry are highly valuable and clear. The results presented also highlight the clear understanding of current scholarly engagement in terms of the role of silence in Social Psychology, addressing the need for further studies. My comments below are minor, and I hope some of them are useful. This is mainly to add some relevant references and the structural organization of sections between 12-16.
Section 5 Methodology
In the second stage, the author included the analytical perspectives involved in this study, such as a) content analysis, b)discourse analysis, c) hermeneutic interpretation and d) information extraction. It would be beneficial for the readers to know some references for the approach used a)-c). For example, discourse analysis can be critical discourse analysis when the ideological approach is involved (e.g., Norman Fairclough’s model).
Before the figure is presented, a paragraph explaining the screening procedure can be added. Section 16 can be added here.
Implications and future directions
Further information about a journal mentioned can be added (e.g., when founded and with the hyperlink).
Order of sections: Sections 12-16
To maintain the coherence in presenting the findings of the study. I think that consideration to change the order of sections 12-16 would be helpful.
As mentioned earlier, section 16 (1st paragraph) can move to the last paragraph under 5. Methodology section by removing the current heading. The second paragraph can possibly be used as a general introduction to the analysis section starting before section 6, as a bridge to the in-depth analysis sections or as the summary towards the end of the paper.
The possible order to maintain the logical coherence can be: (This is just a suggestion)
- General interpretations of the results – (The current 2nd paragraph)
- Implications and future findings (the author may consider to change the wording of this heading, as this section should work as the main findings/ summary as a systematic review
12 and 15 to be merged (limitations of study)
and then conclusion
Before the conclusion or within the conclusion, I would encourage the author to add the significance of this study in a more explicit way.
Author Response
Responses to Reviewer 2’s Comments
Thank you very much for your thorough reading of the manuscript and insightful suggestions. I find them particularly useful and have responded with care to each of them. Below are what I have modified, all highlighted in yellow (the blue highlighted words are my responses to the other reviewer).
Recommendation 1: Under Section 5 Methodology, in the second stage, the author included the analytical perspectives involved in this study, such as a) content analysis, b)discourse analysis, c) hermeneutic interpretation and d) information extraction. It would be beneficial for the readers to know some references for the approach used a)-c). For example, discourse analysis can be critical discourse analysis when the ideological approach is involved (e.g., Norman Fairclough’s model).
Response: I have incorporated some references for the approach used, which are on pages 3 and 4. I have also added the relevant references to the list on page 24. For your reference, I also pasted the changes here:
These texts were subsequently analyzed in a second wave using four key strategies: (a) content analysis, wherein patterns over the surface of the material are recognized [112]; (b) discourse analysis, in which attention is paid to how language reflects ideology and power relations [113]; (c) hermeneutic interpretation, based on traditions of meaning-making through engaging textually [114]; and (d) close reading, which involves extracting information at the level of detail. (pp.3-4)
[112] Krippendorff, K. (2018). Content analysis: An introduction to its methodology. Sage publications.
[113] Fairclough, N. (2023). Critical discourse analysis. In The Routledge handbook of discourse analysis (pp. 11-22). Routledge.
[114] Gadamer, H. G., Marshall, D. G., & Weinsheimer, J. (2004). Truth and method: Continuum impacts.
(pp.24)
Recommendation 2: Before the figure on page 5 is presented, a paragraph explaining the screening procedure can be added. Section 16 can be added here.
Response: I have added an explanatory paragraph to make sure the chart is easily understood and well received by the reader, as seen on page 4. I have then named the figure on page 5 accordingly for clarity. For your reference, I also pasted the changes here:
The researcher carried out the systematic review in several stages to ensure this process would be transparent and rigorous as possible. In the first step, all types of records found through database searching were combined and duplicates were deleted. The studies that were left were then screened on titles and abstracts to delete those studies that seemed would definitely be irrelevant to silence in social psychology. In the second stage, full-text articles were reviewed for relevance, methodological integrity and conceptual relevance to the inclusion and exclusion criteria. Studies that did not contribute with empirical or theoretical insights were not included. This process led to the final range of publications that were included in the review as outlined in the flow diagram (Figure 1). (p.4)
Figure 1 – The Screening Process of Academic Resources (p.5)
Recommendation 3: In section implications and future directions, further information about a journal mentioned can be added (e.g., when founded and with the hyperlink).
Response: This is a great suggestion. I have created a hyperlink to this journal and add more information about its foundation. The modified sentence is below:
The Journal of Silence Studies in Education, for instance, founded in Australia in 2021 by experienced scholars in this area, has enumerated eighty distinct themes for future research, hinting at the need for even more exploration.
Recommendation 4:
Regarding the order of sections 12-17, to maintain the coherence in presenting the findings of the study. I think that consideration to change the order of sections 12-16 would be helpful. As mentioned earlier, section 16 (1st paragraph) can move to the last paragraph under 5. Methodology section by removing the current heading. The second paragraph can possibly be used as a general introduction to the analysis section starting before section 6, as a bridge to the in-depth analysis sections or as the summary towards the end of the paper.
Response: I have followed this suggestion and have moved section 16 to the last paragraph under 5. Methodology (which is now on page 4). I have also transferred the second paragraph can possibly be used as a general introduction to the analysis section starting before section 6 (which is now on page 6). I am grateful for this wonderful suggestion, which has made a difference in the coherence of my article.
Recommendation 5: The possible order to maintain the logical coherence can be: (This is just a suggestion). Section 16 ‘Implications and future findings’ (the author may consider to change the wording of this heading, as this section should work as the main findings/ summary as a systematic review.
Response: I’m happy with this suggestion. I have replace the heading ‘Implications and future findings’ with ‘Summary of main findings.’ I wrote the section a new to reflect the whole article. This section is now number 14. ‘Summary of main findings’ on pages 17 ad 18, highlighted in yellow.
Recommendation 6: 12 and 15 to be merged (limitations of study) and then conclusion
Response: You made such a helpful suggestion that helped me tidy up my writing, thanks. I have merged sections 12 and 15 as proposed. These two have been combined into section 12 ‘Limitationsof the study,’ which now sits on page 15. The transferred paragraph is highlighted in yellow.
Recommendation 6: Before the conclusion or within the conclusion, I would encourage the author to add the significance of this study in a more explicit way.
Response: I have done so. I created section ‘15. Significance of the study.’ It is on page 18 and is highlighted. The conclusion has been renumber accordingly.
Besides the above, I have cleaned up all syntactic and lexical issues in the article. The whole manuscript is now completely error-free.
Thank you very much for all your recommendations.

Reviewer 2 Report
Comments and Suggestions for Authors
Dear Authors,
Thank you for the opportunity to review your manuscript. Please refer to the attached file for detailed comments and suggestions. I hope the feedback will be helpful for your revision.

Author Response
Responses to Reviewer 1’s Comments
Thank you very much for your insightful comments. The work has been thoroghly revised with great care to repond to your suggestions. Below is the detailed explanations, while all the evidence of the revised words are now highlight in blue colour throughout the article (while the yellow highlighted words are my responses to the second reviewer).
Recommendation 1a: Although the author claims to follow the PRIMA (presumably PRISMA) guidelines, the paper lacks essential elements such as a flow diagram outlining the screening process, a detailed table of inclusion and exclusion criteria, and a quality appraisal of the reviewed literature.
Response: A flow diagram is presented on page 4 of the manuscript. Thank you.
Recommendation 1b: Furthermore, the use of a fully manual and intuitive approach to article selection raises concerns about the scientific rigor and replicability expected in systematic reviews.
Response: This issue has been well taken care of in three ways.
First, the artcile selection process was made that it relies on 5 criteria. As stated on page 3, which refers to ‘the inclusion of silence studies (a) empirical research, (b) historical perspectives, (c) recent developments, (d) relevance to current discussions, and (e) comprehensive debates.’
Second, this limitation is acknowledged in section 12 (Limitations of the study) on page 12, which states that ‘each selected article reflects the unique perspective of its author, and thus may not encapsulate a comprehensive view of silence studies across the academic community.’
Third, I have also added 2 last paragraphs at the end of page 13 to provide specific details in the data selection process. This part is highlighted in yellow since both reviewers had suggested for the same modification. I also attached the added part here for your reading convenience:
The initial search yielded 400 records across multiple academic databases, including Scopus, JSTOR, PsycINFO, and Google Scholar, spanning research published from the 1970s to 2025. After removing duplicates and non-English texts, 342 articles remained for preliminary screening. Titles and abstracts were reviewed for relevance to the social psychology of silence, leading to the exclusion of 189 records. Full texts of the remaining 153 articles were assessed using predefined inclusion criteria: relevance to silence as a psychological or sociocultural construct, conceptual clarity, and contextual diversity. Ultimately, 111 studies were included in the final synthesis.
The review highlights silence as a multidimensional construct that transcends cultural, psychological, and communicative boundaries. The included studies show that silence can be both deliberate and unconscious, functioning as a form of resistance, contemplation, politeness, marginalization, or emotional regulation. A dominant theme across the literature is the cultural relativity of silence, with varying norms and interpretations found across East Asian, Western, and Indigenous contexts. For instance, what may be perceived as passivity in one context may represent thoughtfulness or deference in another.
Recommendation 2: The manuscript does not provide any statistical summary of the reviewed sources in terms of their publication period, regional focus, or research type. The absence of such categorization weakens the clarity and analytical depth of the synthesis and limits the potential for identifying research trends or gaps.
Response: Thank you for this request. Statistical summaries of the reviewed sources have been provided in the following two specific locations in the article:
A mention of the statistical 110 texts can be found on page 3.
There is also an inclusion on page 3 of the 400 academic texts concerning silence were chosen from a wide range of databases including PubMed, Scopus, Web of Science, JSTOR, Academic Search Complete, ScienceDirect, Google Scholar, Cochrane Library, PsycINFO, ERIC, ProQuest, IEEE Xplore, and SAGE Journals.
Recommendation 3: The overall structure of the paper lacks coherence. There is a notable disconnect between sections-for example, "silence as kindness" and "silence as trauma" are juxtaposed without a clear theoretical or conceptual bridge. Additionally, much of the content consists of lengthy descriptive excerpts rather than critical analysis. This is especially evident in the discussion of cultural differences, where numerous cases are presented without synthesizing them into a cohesive theoretical framework.
Response: A clear paragraph has been added to pages 11 and 12 that clarifies the foundation that connect the seemingly unassociated concepted about silence. The added segment stated this:
Though these views on silence may appear to run counter, they are linked by the same foundation of attention towards how silence is inundated with relational significance. Silence, socially and psychosocially, is a relational act, that can either grant dignity or violate it. But in the hands of writers, silence can also serve as an intentional communicator of care, respect or forgiveness that opens up spaces of empathy and recognition. But when the silence is forced or born from fear, shame, or exclusion, that absence of words can compound alienation and cause harm. In so doing, silence is not inherently positive or negative in and of itself: it is an issue of relationality, and meaning depends on both the circumstances, power, and the quality of the human relationship. In holding these dualities together, we also see silence as not a pair of opposites but a spectrum of human experience — able to envelop both kindness and trauma, with countless gray shades in between.
Recommendation 4: Although the paper references various theoretical perspectives from sociocultural studies, cognitive psychology, and linguistics, it does not adopt a clearly articulated theoretical framework to anchor the analysis (e.g., Goffman 's frame theory or Bourdieu 's theory of symbolic capital). As a result, the discussion remains somewhat fragmented, and the interpretive depth is limited.
Response: Thanks for sharing which is raising awareness about theorization. I have taken in depth into the above theory, and in thought, have realized by a thorough analysis that although these perspectives are interesting, it would not work to map the variety of existing scholarship on silence for the sake of their interpretation through one theoretical lens. Such an anchoring of the analysis in a single framework could with revised permission have an unintended side effect of ‘fencing in’ the analysis, especially since the studies under scrutiny come from a range of academic traditions including psychology, sociolinguistics, anthropology, and education. Because of this, I have chosen a pluralistic methodology that brings to the fore the multi-dimensionality of silence in various contexts. I think this approach captures best the heterogeneity of the literature and translates well the disparate insights provided from various disciplinary vantage points, but still allows us to draw synthetic links across them. Thanks for the opportunity to think a bit more coherently about this theoretical field.
Recommendation 5: While the conclusion summarizes multiple dimensions of silence (e.g., linguistic, cultural, cognitive), it does not clearly articulate the paper's theoretical contributions or practical implications. A systematic review should offer more than a descriptive overview; it should identify how the current work integrates or extends existing knowledge, and propose a research agenda based on the identified gaps.
Response: Yours is a valuable suggestion that really helps me enhance the work. I have thus added the following paragraph to the conclusion on page 15:
Markedly, this review adds to the existing literature theoretically by considering silence not as a marginal or leftover category of communication, but rather as a multidimensional construct cutting across cognitive, sociocultural, and psychological domains. By comparing insights from more than 100 studies this chapter adds to current understanding and illustrates how silence operates at once as an intrapersonal resource (for reflection and self-talk), an interpersonal strategy (for being caring or resistant or negotiative), and a sociopolitical marker (of power and exclusion and trauma). These theoretically driven findings have practical implications to indicate that educators, health professionals and individuals in policy roles need to reconceptualize silence as an active and productive state of participation, as opposed to an absence. Further study could extend this synthesis to an examination of how silence functions within digital spaces, with marginalized groupings, and among the interculturally diverse to facilitate theoretical grounding for future work on understanding communication complexity.
Besides the above, I have cleaned up all syntactic and lexical issues in the article. The whole manuscript is now completely error-free.
Thank you very much for reading my manuscript and for providing such useful feedback that really assists me in this learning experience.

Round 2
Reviewer 2 Report
Comments and Suggestions for Authors
I appreciate the authors’ efforts in revising the manuscript “A Systematic Review of Multifaceted Silence in Social Psychology.” The revised version shows significant improvements in terms of structure, methodological transparency, and theoretical articulation. The inclusion of a PRISMA-like flow diagram, explicit inclusion/exclusion criteria, and thematic synthesis represents clear progress compared to the initial submission. That said, while the revisions are commendable, several issues remain that need further attention before the paper can fully meet the standards of a systematic review in the field.
While the addition of screening criteria and a flow diagram is a step forward, the paper still lacks a systematic quality appraisal of the included studies. Without such an assessment, the review risks remaining largely descriptive rather than meeting the standards of a rigorous systematic review. In addition, although a bar chart on thematic distribution has been added, the manuscript does not provide sufficient quantitative summaries of the included studies by period, region, or research method, which limits its ability to reveal disciplinary trends and research gaps.
In terms of structure, the revised organization around “interpersonal silence” and “intrapersonal silence” has improved coherence, and the attempt to conceptualize silence as a spectrum is a positive step. However, some sections remain overly descriptive, relying heavily on examples, and the critical synthesis could be deepened. Theoretically, the introduction of Mills’ notion of the sociological imagination provides a stronger perspective, but the discussion still draws on multiple frameworks without fully consolidating them into a coherent theoretical anchor.
Furthermore, the conclusion section now articulates practical implications more clearly and outlines directions for future research, but the theoretical contribution remains somewhat modest. The paper integrates and synthesizes existing literature effectively, yet it does not present a new conceptual model or framework that would substantially advance theory.
Overall, the manuscript demonstrates clear progress and is moving in the right direction, but in its current form it still requires further work to meet the standards of a systematic review. A more explicit quality appraisal, more detailed quantitative summaries, and a more consolidated theoretical framework would considerably strengthen the contribution. I therefore recommend that the manuscript undergo another revision.
Author Response
Please see the attached file, thank you.
